# Epidemiological Investigation of Hospital Transmission of *Corynebacterium striatum* Infection by Core Genome Multilocus Sequence Typing Approach

Yutong Kang,[a] Shenglin Chen,[a,d] Beijia Zheng,[b,f] Xiaoli Du,[a] Zhenpeng Li,[a] Zhizhou Tan,[a] Haijian Zhou,[a] Jia Huang,[g] Leihao Tian,[b,c] Jiaxin Zhong,[a] Xueli Ma,[b,c] Fang Li,[e] Jiang Yao,[a] Yu Wang,[h] Meiqin Zheng,[b,c] Zhenjun Li[a]

aState Key Laboratory for Infectious Disease Prevention and Control, National Institute for Communicable Disease Control and Prevention, Chinese Center for Disease Control and Prevention, Beijing, China

bEye Hospital and School of Ophthalmology and Optometry, Wenzhou Medical University, Wenzhou, Zhejiang, China

cNational Clinical Research Center for Ocular Diseases, Wenzhou, Zhejiang, China

dSchool of Public Health, Shanxi Medical University, Taiyuan, Shanxi, China

eDepartment of Medicine, Tibet University, Lhasa, Tibet, China

fDepartment of Clinical Laboratory, Taizhou Central Hospital (Taizhou University Hospital), Taizhou, Zhejiang, China

gInstitute for the Prevention and Control of Infectious Diseases, Xinjiang Center for Disease Control and Prevention, Urumqi, Xinjiang, China

hDepartment of Clinical Laboratory Medicine, Shanxi Bethune Hospital & Shanxi Academy of Medical Sciences, Tongji Shanxi Hospital, Third Hospital of Shanxi Medical University, Taiyuan, Shanxi, China

**ABSTRACT** *Corynebacterium striatum* has recently received increasing attention due to its multiple antimicrobial resistances and its role as an invasive infection/outbreak agent. Recently, whole-genome sequencing (WGS)-based core genome multilocus sequence typing (cgMLST) has been used in epidemiological studies of specific human pathogens. However, this method has not been reported in studies of *C. striatum*. In this work, we aim to propose a cgMLST scheme for *C. striatum*. All publicly available *C. striatum* genomes, 30 *C. striatum* strains isolated from the same hospital, and 1 epidemiologically unrelated outgroup *C. striatum* strain were used to establish a cgMLST scheme targeting 1,795 genes (hereinafter referred to as 1,795-cgMLST). The genotyping results of cgMLST showed good congruence with core genome-based single-nucleotide polymorphism typing in terms of tree topology. In addition, the cgMLST provided a greater discrimination than the MLST method based on 6 housekeeping genes (*gyrA*, *gyrB*, *hsp65*, *rpoB*, *secA1*, and *sodA*). We established a clonal group (CG) threshold based on 104 allelic differences; a total of 56 CGs were identified from among 263 *C. striatum* strains. We also defined an outbreak threshold based on seven allelic differences that is capable of identifying closely related isolates that could give clues on hospital transmission. According to the results of analysis of drug-resistant genes and virulence genes, we identified CG4, CG5, CG26, CG28, and CG55 as potentially hypervirulent and multidrug-resistant CGs of *C. striatum*. This study provides valuable genomic epidemiological data on the diversity, resistance, and virulence profiles of this potentially pathogenic microorganism.

**IMPORTANCE** Recently, WGS of many human and animal pathogens has been successfully used to investigate microbial outbreaks. The cgMLST schema are powerful genotyping tools that can be used to investigate potential epidemics and provide classification of the strains precise and reliable. In this study, we proposed the development of a cgMLST typing scheme for *C. striatum*, and then we evaluated this scheme for its applicability to hospital transmission investigations. This report describes the first cgMLST schema for *C. striatum*. The analysis of hospital transmission of *C. striatum* based on cgMLST methods has important clinical epidemiological significance for improving nosocomial infection monitoring of *C. striatum* and in-depth understanding of its nosocomial transmission routes.

Address correspondence to Zhenjun Li, lizhenjun@icdc.cn, or Meiqin Zheng, zmq@eye.ac.cn.

The authors declare no conflict of interest.

[This article was published on 20 December 2022 with errors in the affiliations. The affiliations were corrected in the current version, posted on 4 January 2023.]

**KEYWORDS** *Corynebacterium striatum*, core genome multilocus sequence typing, clonal groups, outbreak investigation, virulence, resistance

*Corynebacterium striatum* is part of microbiota of skin and nasal mucous membranes of humans and has been increasingly reported as the etiologic agent of hospital- and community-acquired infections (1). Although the significance and prevalence of *C. striatum* as a causative pathogen are not yet fully understood, a growing number of *C. striatum* isolates have been shown to be associated with a wide range of diseases or conditions, including septicemia (2), pulmonary infection (3), meningitis (4), endocarditis (5), osteomyelitis (6), septic arthritis (7), skin wounds (8), intrauterine infections (9), etc. Timely diagnosis and prompt treatment of the infection could lead to a favorable outcome for the patient (10). Several outbreaks of nosocomial infections associated with *C. striatum* have been reported (3, 11–14). Recently, the possibility of patient-to-patient transmission of *C. striatum* has been demonstrated (11, 12, 15). Multidrug-resistant and pathogenic *C. striatum* isolates have been detected in recent years, but the specific clonal groups (CGs) corresponding to these high-risk strains remain unknown. Resistance genes, virulence genes, and predominant clonal group spreading in the hospital environment can be revealed via genomic analysis (16–18). Molecular epidemiological studies and surveillance of *C. striatum* are highly significant.

Currently, pulsed-field gel electrophoresis (PFGE) (14, 15, 19) and multilocus sequence typing (MLST) (20–22) are two major subtyping techniques that are used to identify clonal relationships between *C. striatum* isolates. PFGE is one of the commonly used DNA fingerprinting techniques, but it is time-intensive and difficult to apply to differentiation between closely related bacterial strains. In contrast, MLST, developed in 2012 for *C. striatum*, is based on the sequencing and evaluation of the allelic variation of seven housekeeping genes, and it characterizes isolates by generating so-called sequence types (STs) (20). To offer an enhanced level of resolution, MLST is typically used alongside PFGE (23, 24). High-throughput next-generation sequencing technologies are expected to revolutionize medical microbiology and molecular epidemiology (25, 26) by improving the discriminatory power of molecular typing and determining the resistance and virulence of clinical isolates. However, extracting medically relevant information from genome sequences remains challenging. In recent years, core genome MLST (cgMLST) has become increasingly popular for epidemiological research and subspecies-level classification (27–30). cgMLST is for expansion of the MLST concept to a greater number of genes of the core genome and aims to combine the discriminatory power of classical MLST with the extensive genetic data obtained by whole-genome sequencing. The goal of this study was to develop a cgMLST scheme and delineate precisely, based on a cgMLST strategy, CGs corresponding to highly virulent and multidrug-resistant *C. striatum* isolates and investigate potential hospital transmission of *C. striatum*.

## RESULTS

***C. striatum* cgMLST scheme.** For the scheme creation (Fig. 1), the 271 genome sequences were used, and using WP1a as the reference genome, a total of 4,934 target loci were annotated to generate the whole-genome MLST (wgMLST) data set. In the filtering steps, 66 paralogous loci were discarded, and genome quality testing filtered out an additional 4,868 locus targets. A total of 1,917 gene targets were retained as candidates for the cgMLST scheme. For the evaluation step, 31 *C. striatum* genomes were collected from two hospitals, thoroughly filtered following the same steps used for the scheme creation, and analyzed using the candidate list of 1,917 gene targets. Following this step, a cgMLST scheme composed of 1,795 gene targets (referred to here as the 1,795-cgMLST scheme) was defined, covering 67.1% of the 2,676 open reading frames predicted for the reference strain, WP1a.

**Definition of CGs based on cgMLST.** To avoid defining CGs arbitrarily, we analyzed the distribution of the number of allelic differences among all of the pairs of genomes. The results demonstrated a high number of genome pairs with <104 or >650 allelic differences; almost no genome pairs had 104 to 650 allelic differences (Fig. 2). On the

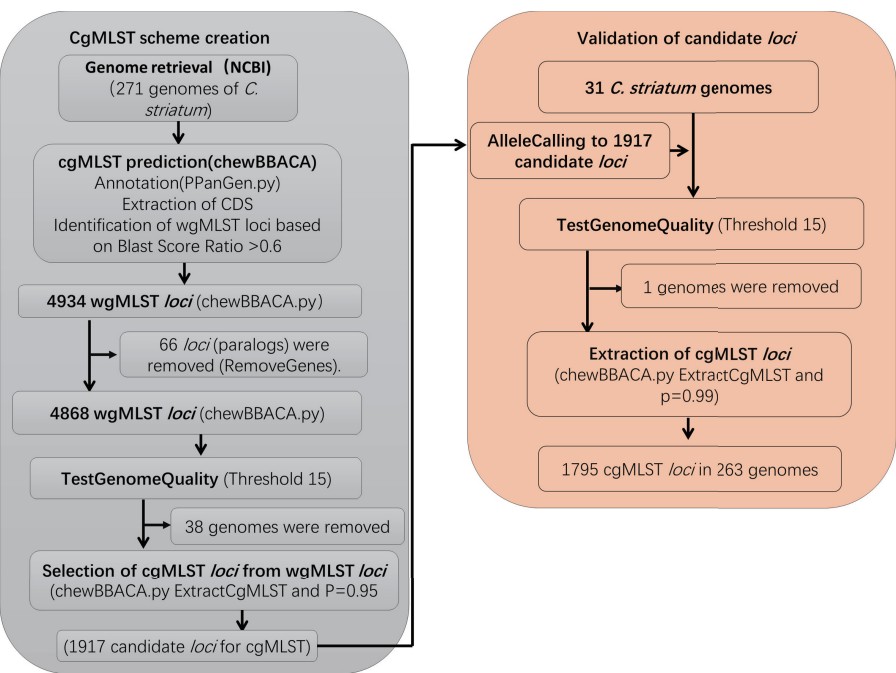

**FIG 1** Flowchart to determine cgMLST scheme for *C. striatum* using chewBBACA (https://github.com/B-UMMI/chewBBACA).

basis of the above-described analysis, a threshold of 104 allelic differences was chosen as the cutoff score to define CGs. Consequently, a CG was defined as a group of cgMLST allelic profiles differing by <104 allelic mismatches, out of 1,795 gene loci, from at least one other member of the group. Similar to classical MLST data, cgMLST data can also be used to devise a classification of strains using the single-linkage algorithm (31). Therefore, CGs were identified based on single-linkage clustering of cgMLST allelic profiles with the allelic difference threshold of 104. In total, 56 CGs were identified by using this definition (see Table S3 in the supplemental material).

The minimum spanning tree generated with the CGs detected is presented in Fig. 3. *C. striatum* strains that belonged to different CGs could be distinctly clustered. CG3 was the most common among all of the *C. striatum* strains in this study (32.3%). In addition to CG3, we identified high rates of CG4 (16.0%), followed by CG5 (5.3%), CG2 (4.9%), CG6 (4.6%), CG19 (3.4%), CG18 (3.0%), CG12 (2.3%), CG7 (1.9%), CG11, CG14, and CG21 (four isolates each), CG9, CG10, and CG20 (three isolates each), CG1, CG8, CG13, CG15, CG16, CG17, and CG22 (two isolates each), and CG23 to CG56 (one isolate each).

Geographically, strains were mainly isolated from four countries (China, 246; United States, 11; Germany, 2; Denmark, 1; and unknown, 3) (Fig. 4a). CG3 and CG12 were shared by Beijing, Tangshan, and Guangdong, China, while CG3, CG4, CG5, CG6, CG9, CG11, CG12, and CG14 were shared only by Beijing and Tangshan. The other 46 CGs showed a clear regional distribution. There was a pronounced correlation between locations and some specific CGs. CG1, CG23, CG24, CG25, CG26, CG27, CG45, CG46, CG47, and CG56 were prevalent exclusively within the United States. CG48 and CG49 were prevalent exclusively within Germany. CG50 was prevalent exclusively within Denmark. CG52 was prevalent exclusively within Wenzhou, China. CG18, CG19, CG20, CG21, CG22, CG53, CG54, and CG55 were prevalent exclusively within Shanxi, China. CG7, CG8, CG13, CG30, and CG31 were prevalent exclusively within Tangshan, China. CG28, CG32, CG33, and CG34 were prevalent exclusively within Guangdong, China. CG2, CG10, CG16, CG17, CG29, CG35, CG36, CG37, CG38, CG39, CG40, CG41, CG42, CG43, and CG44 were prevalent exclusively within Beijing, China (Fig. 4b).

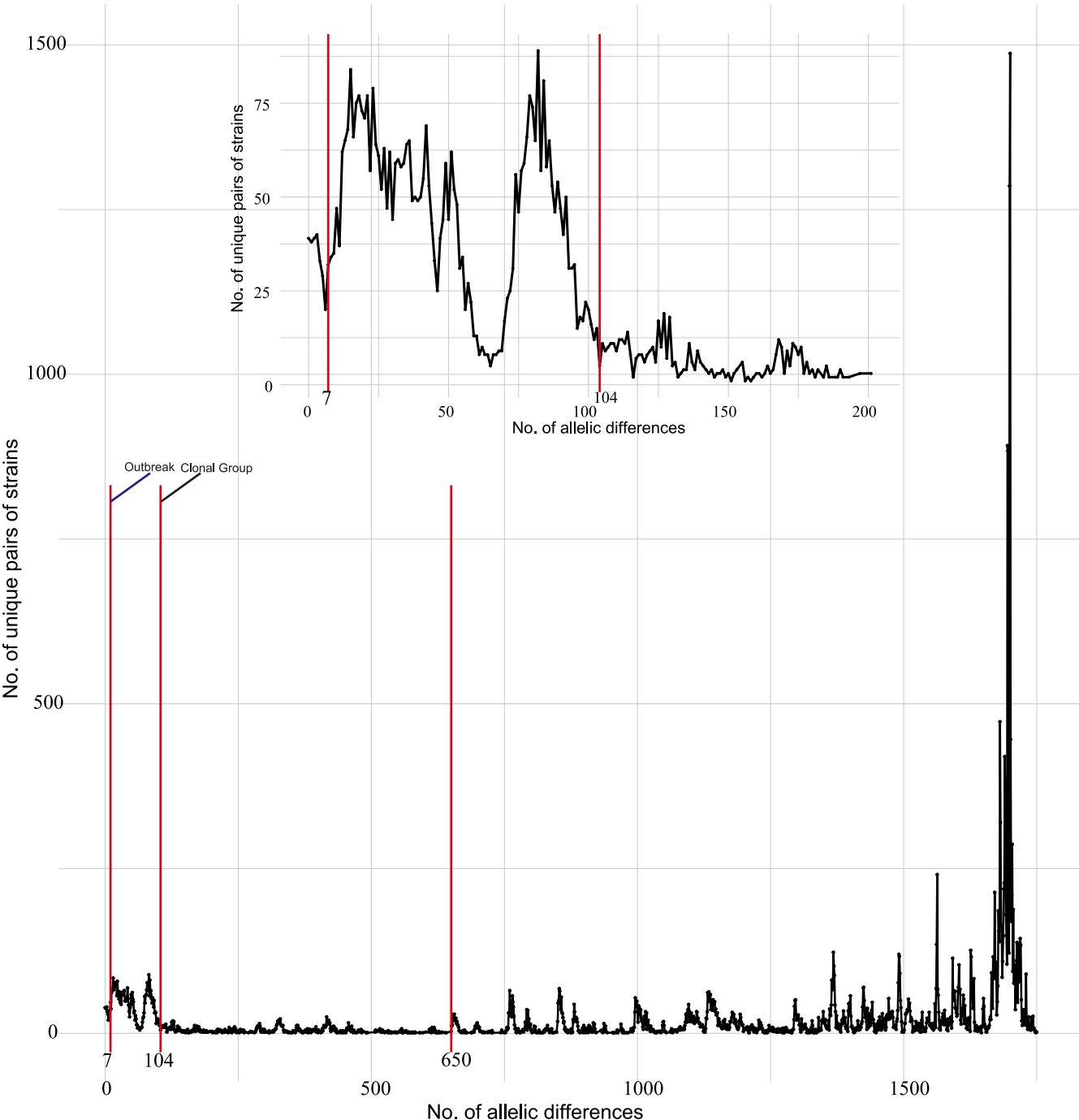

**FIG 2** Distribution of the number of pairwise allelic mismatches, which is the number of loci at which sequences of a given pair of strains differ. An allelic mismatch cutoff of 104 is recommended to define the clonal group, which is represented in red. An allelic mismatch cutoff of 7 is recommended to define the outbreak threshold, which is represented in red.

**Comparison of cgMLST and MLST.** MLST loci in general should represent genes encoding proteins with functions (housekeeping genes) because they are considered more stable in terms of genetic mutation (32). Maiden recommended that an MLST scheme include 6 to 10 loci (33). Previous studies employed 3 different combinations of housekeeping genes to determine the sequence types (STs), including (i) *ITS1*, *gyrA*, and *rpoB*, (ii) 16S rRNA, *ITS1*, *gyrA*, and *rpoB*, and (iii) *sodA*, *rpoB*, *gyrA*, *gyrB*, *hsp65*, *secA1*, and 16S rRNA. Nevertheless, an MLST typing scheme for *C. striatum* is currently not publicly available. Considering the conservation and protein-coding function of

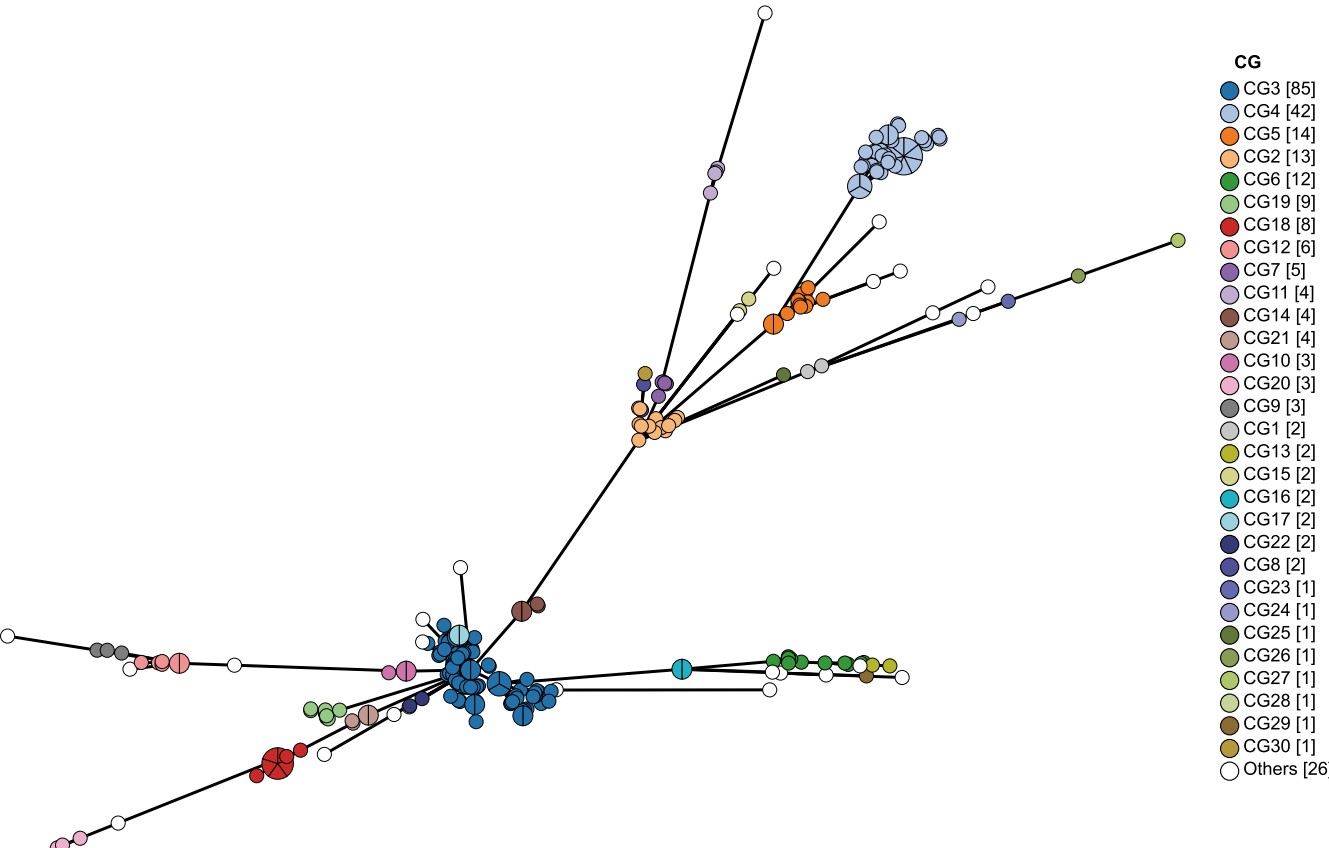

**FIG 3** Minimum spanning tree based on cgMLST allelic profiles of 263 *C. striatum* isolates. Nodes corresponding to a unique allelic profile are colored according to their corresponding CGs.

housekeeping genes, we selected 6 housekeeping genes (*gyrA*, *gyrB*, *hsp65*, *rpoB*, *secA1*, and *sodA*) to determine the STs. The *gyrA*, *gyrB*, *hsp65*, *rpoB*, *secA1*, and *sodA* genes had 20, 22, 25, 17, 16, and 12 polymorphic sites, respectively, and the average numbers of nucleotide differences were 1.48, 6.402, 4.246, 2.911, 1.813, and 3.262, respectively (Table S4). Using the six genes, 38 distinct STs were identified (Table S5), whereas cgMLST distinguished 56 CGs. ST33 (33.5%) was the most abundant, followed by ST25 (16.0%). Consistent with this, MLST did not subtype any CG, whereas multiple STs could be subdivided using cgMLST (Fig. S3). This indicates that cgMLST improves discrimination among *C. striatum* isolates compared with MLST.

**Comparison between cgMLST cluster analysis and SNP-based phylogeny.** To compare the typing results of cgMLST and sequence-based methods, we extracted the nucleotide sequences of the 1,795 cgMLST target genes from 263 genomes included in the cgMLST scheme. Genomic comparison revealed 663,718 core genome single-nucleotide polymorphisms (SNPs) among all 263 isolates. Sequence-based phylogenetic analysis of core polymorphic sites was performed and compared with the clustering based on cgMLST allelic profiles. A tanglegram plot was generated for a visual comparison of cgMLST and SNPs, and it showed good congruence between the two methods (Fig. S4). There were some subtle differences in the two phylogenies, mainly caused by inversions of clusters. This was explained by differences between the internal nodes located deeper within the phylogenies.

**Retrospective analysis of nosocomial infection outbreak strains.** In a hospital in Shanxi, China, in-hospital prevalence of *C. striatum* infection was relatively stable each month. About 30 to 40 strains can be isolated every month. In our present study, we collected all strains isolated from 24 March to 23 April 2021 for whole-genome sequencing to validate our cgMLST scheme. A total of 30 *C. striatum* isolates were obtained from

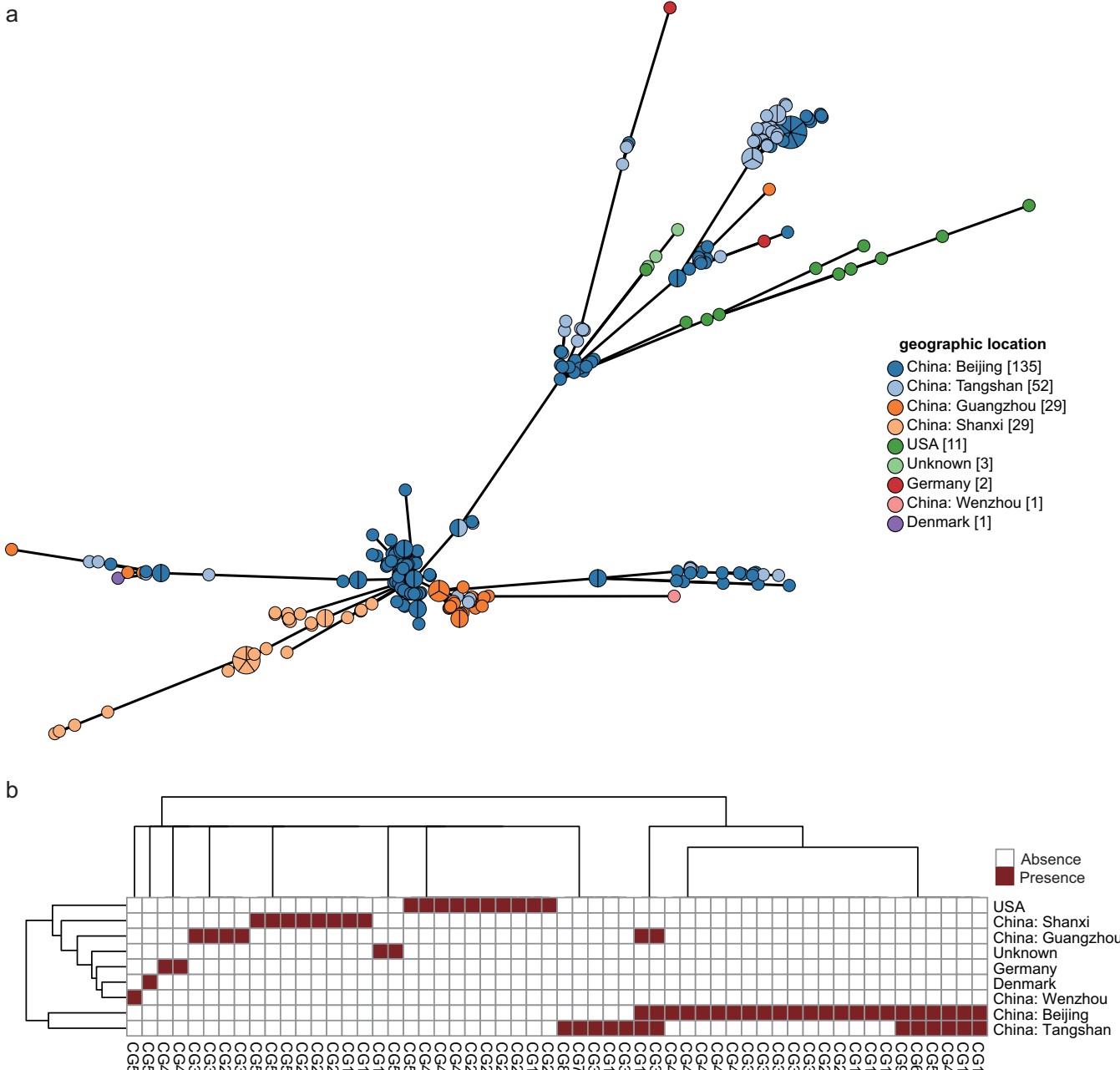

**FIG 4** (a) Minimum spanning tree based on cgMLST allelic profiles of 263 *C. striatum* isolates. Nodes corresponding to a unique allelic profile are colored according to their corresponding locations of origin. (b) Heat map of the distribution of CGs in different countries.

26 patients (Fig. 5a). Two of the 26 patients had been hospitalized for lung infections. The remaining 24 patients had been hospitalized for some underlying condition and contracted *C. striatum* during their stay. Two or more times cultures were positive for 3 patients. Among the 30 *C. striatum* isolates, 7 isolates originated from tracheal secretions, 22 isolates were from sputum, and 1 isolate was from a catheter. *C. striatum* was obtained as a pure culture growth from these clinical specimens, and the patients had symptoms of infection; we believe that *C. striatum* isolated from all patients was the pathogen of infection, not a colonizer. Since isolate 34 was deleted during the validation step, only 29 strains isolated from Shanxi and 1 epidemiologically unrelated strain isolated from Wenzhou were included in the subsequent analysis (Fig. S5a and Table S6).

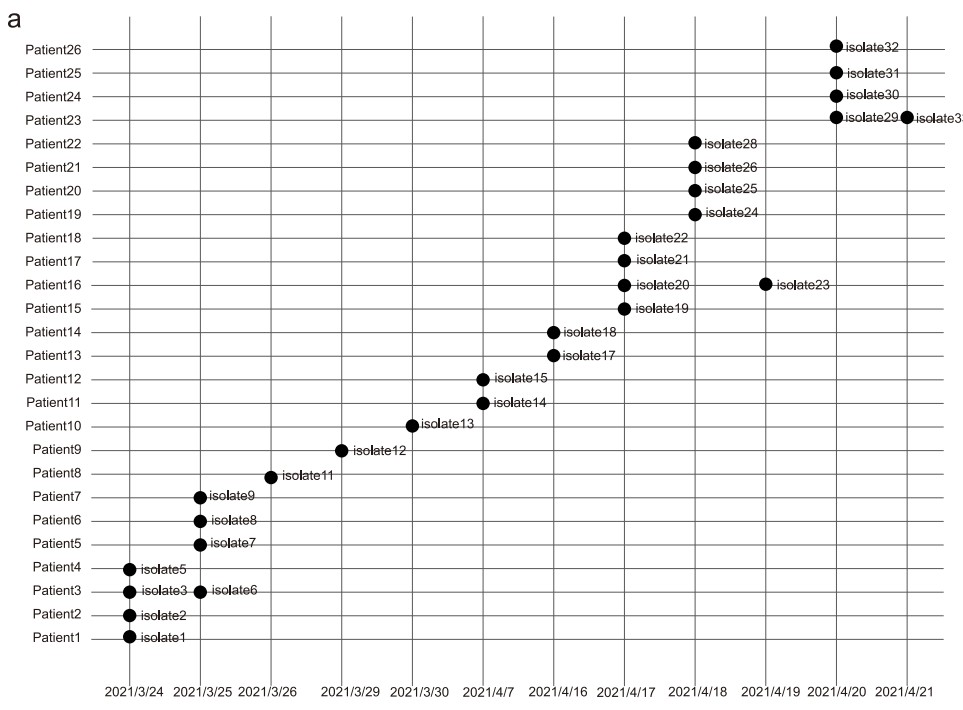

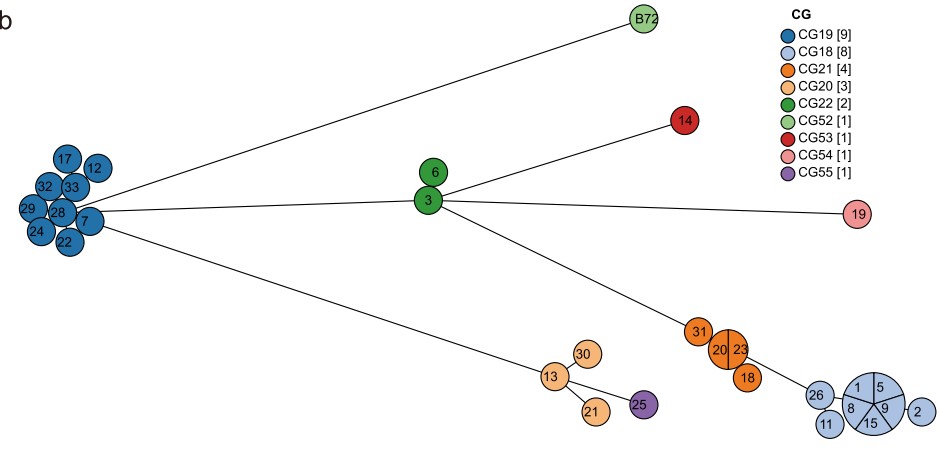

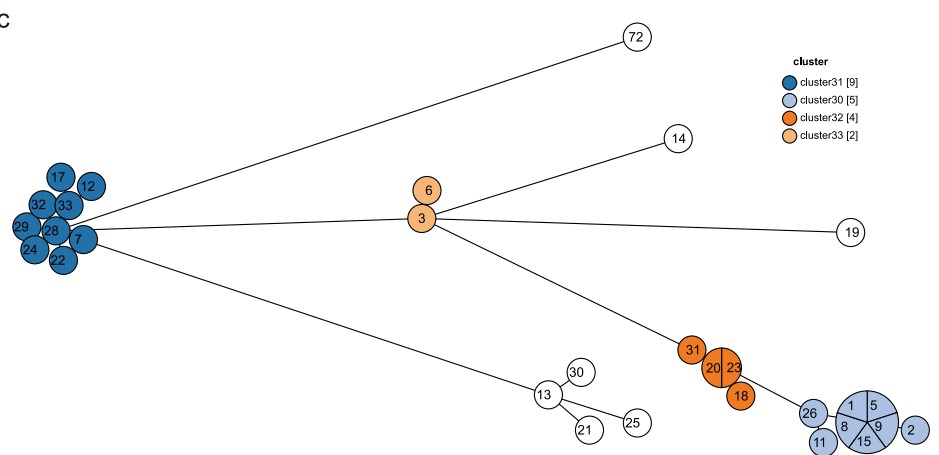

**FIG 5** (a) Grid plot for the isolation time and patients for all *C. striatum* strains. Dots with different colors represent different transmission clusters. (b and c) Minimum spanning trees based on cgMLST allelic profiles of 30 validation *C. striatum* isolates. Nodes corresponding to unique allelic profiles are colored according to their corresponding CGs (b) and hospital transmission clusters (c).

Of the isolates isolated in Shanxi, 19 were from the Department of Neurosurgery. The remaining were sporadically distributed among 6 other departments, including the Department of Respiratory and Critical Care Medicine (2 isolates), Department of Integrated Healthcare (1 isolate), Department of Rehabilitation Medicine (2 isolates), Department of Critical Care Medicine (3 isolates), Department of Cardiovascular Internal Medicine (1 isolate), and Department of Neurology (1 isolate) (Fig. S5b). These units were housed in one surgical building and one internal medicine building. The Department of Neurosurgery and Department of Critical Care Medicine were housed in surgical building; the other departments were housed in the internal medicine building. Twenty-three isolates were from the surgical building, and 7 isolates were from the internal medicine building (Fig. S5c). The surgical building contained six CGs (CG18, CG19, CG20, G21, CG22, and CG55); the internal medicine building also contained six CGs (CG18, CG19, CG20, CG21, CG53, and CG54). Main CGs (CG16, CG17, and CG19) were shared by both buildings (Fig. 5b and Fig. S5b). There was no pronounced correlation between the hospital units and CGs.

The results of Wang et al. also demonstrated that *C. striatum* has spread in parallel across China, causing persistent and extensive transmissions within hospitals (34). Therefore, it is necessary to define an "outbreak threshold" based on cgMLST to identify strains from the same outbreak and closely related isolates and to investigate the nosocomial spread of *C. striatum*. According to previous research on outbreak investigation based on cgMLST, the number of allelic differences among outbreak-related strains is generally less than 40, as isolates from the same outbreak are very closely related (35–41). Therefore, the clustering efficiency of 1 and 40 allelic differences is measured by calculating the Dunn index (DI) (42) (Fig. S6). Since cluster distances are measured by allelic differences, the network with the best clustering efficiency (i.e., the highest DI) will also produce clusters that are most representative of biological relationships, as isolates within clusters are more closely associated with each other than with isolates from other clusters. There is sharp discontinuity at 3 to 10 allelic differences (Fig. 2). A value of 7 for allelic differences has the best clustering efficiency (Fig. S6), a clear local maximum, making this cutoff selected as an outbreak threshold. Therefore, we propose a threshold of 7 for clustering and indicating "possible" transmission.

Twenty-seven *C. striatum* strains from patients with hospital-acquired infections, 2 *C. striatum* strains from patients with community-acquired infections, and 1 epidemiologically unrelated outgroup *C. striatum* strain were used to validate the ability of our cgMLST scheme to investigate hospital transmission of *C. striatum*. There were four transmission clusters with 2 or more isolates differing by ≤7 allelic differences in the Shanxi hospital (Fig. 5c). Of note, strains in transmission cluster 31 and cluster 32 were isolated from patients in different departments and buildings, suggesting the interunit and interbuilding spread of *C. striatum* in the hospital. Most of the transmissions occurred within the Department of Neurosurgery and in agreement with the findings by Wang et al. (34). Isolate 14 and isolate 19 from patients with community-acquired infections and epidemiologically unrelated isolate 72 formed three separate clonotypes and transmission clusters. This shows that CG classification could possibly inform on the epidemiological links among *C. striatum* isolates.

Some *C. striatum* isolates from the same patient had the same CG patterns and belonged to the same transmission cluster. Isolate 3 (isolation date [year/month/day], 2021/3/24) and isolate 6 (2021/3/25), isolated from patient 3, belonged to CG22 (cluster 33). Isolate 20 (isolation date, 2021/4/17) and isolate 23 (2021/4/19), isolated from patient 16, belonged to CG21 (cluster 32). However, isolate 29 (isolation date, 2021/4/20) and isolate 33 (2021/4/21), isolated from patient 23, belonged to CG1 (cluster 31). The 1,795-cgMLST scheme provides high-level resolution, allowing discrimination among closely related *C. striatum* strains.

To evaluate the ability of the scheme to analyze nosocomial transmission, we reinvestigated 213 published nosocomial-transmission strains isolated from three hospitals in three regions of China included in our cgMLST scheme (34). We used ≤7 allelic

differences to infer transmission events. The hospital transmission events occurred in all three hospitals. A total of 28 transmission clusters were discovered: 8 in the hospital in Hebei, 15 in the hospital in Beijing, and 5 in the hospital in Guangdong. The strains within each transmission cluster were isolated in the same hospital; potential interhospital transmission events were not found. The strains within transmission cluster 1, cluster 5, cluster 6, cluster 7, cluster 8, cluster 16, cluster 19, and cluster 25 were isolated in the same units. The strains within the remaining transmission clusters were isolated in different units. In the previous study, in addition to cluster 9 and cluster 13, some or all of the isolates of the remaining 26 clusters were also identified as hospital transmission-related strains by using ≤2 SNP differences (Table S7).

**Presence and diversity of resistance genes among *C. striatum* strains.** To determine the resistome, we conducted *in silico* analysis to identify genes associated with antibiotic resistance by screening each *C. striatum* genome with the ABRicate pipeline using the CARD database. We found that 218 *C. striatum* isolates carried two or more antimicrobial resistance genes (Fig. 6). In contrast, 9 isolates were entirely devoid of resistance genes. A high proportion of *C. striatum* strains (94.7%) harbored the *ErmX* gene, which is associated with streptogramin antibiotics, lincosamide antibiotics, and macrolide resistance. The *sul1* gene, which confers resistance to sulfonamide antibiotics, was present in 68.1% of *C. striatum* strains. The *tetW* gene, which confers tetracycline resistance, was found in 65.0% of *C. striatum* strains. In contrast, much lower percentages of the strains carried resistance markers for other antimicrobials. The *aac(6′)-IIa*, *aac(6′)-Ib7*, and *aph(3′)-VIa* genes, which confer aminoglycoside resistance, were detectable in 1 strain, 1 strain, and 4 strains, respectively. The *TEM-181* gene, which confers phenicol, diaminopyrimidine, and fluoroquinolone resistance, was found in 1 strain. The *DfrA37* gene, which confers penam, penem, monobactam, and cephalosporin resistance, was found in 1 strain.

It is noteworthy that the distribution of antimicrobial resistance genes was mainly associated with particular CGs. CG26 and CG55 carried the highest number of antimicrobial resistance genes (11). Congruently, CG28 and 12 isolates of CG5 (85.7%) carried 10 antimicrobial resistance genes, namely, the *ErmX*, *sul1*, *qacEdelta1*, *tetW*, *cmx*, *aph(3′)-Ia*, *APH(3″)-Ib*, *tetA*, *aph(6)-Id*, and *aac(6′)-Ib10* genes. Congruently, 24 isolates of CG4 (57.1%) carried 9 antimicrobial resistance genes [*ErmX*, *sul1*, *qacEdelta1*, *cmx*, *aph(3′)-Ia*, *aph(3″)-Ib*, *tetA*, *aph(6)-Id*, and *aac(6′)-Ib10*] and 16 isolates of CG4 (38.1%) carried 8 antimicrobial resistance genes [*ErmX*, *sul1*, *qacEdelta1*, *cmx*, *aph(3′)-Ia*, *aph(3″)-Ib*, *tetA*, and *aac(6′)-Ib10*]. Similarly, all isolates of CG14, CG17, CG18, CG19, CG22, CG34, CG37, CG38, CG41, and CG53 and almost all isolates of CG3 (98.8%) harbored five resistance genes, namely, the *ErmX*, *sul1*, *qacEdelta1*, *tetW*, and *aac(6′)-Ia* genes. All of the isolates of CG9, CG10, CG12, CG26, CG29, CG31, CG35, CG40, CG42, CG44, CG48, CG50, and CG54 harbored only the *ErmX* genes. All of the isolates of CG15, CG30, CG43, CG49, CG7, and CG8 and 7 isolates of CG2 (53.8%) harbored the *ErmX*, *tetW*, *cmx*, *aph(3′)-Ia*, *aph(3″)-Ib*, *tetA*, and *aph(6)-Id* genes.

Among all 263 isolates, antibiotic resistance phenotype data of 213 isolates were obtained from Wang et al. (34), including for 12 antimicrobial agents, namely, gentamicin (GEN), penicillin (PEN), meropenem (MEM), cefotaxime (CTX), erythromycin (ERY), clindamycin (CLI), tetracycline (TET), doxycycline (DOX), linezolid (LZD), rifampin (RIF), ciprofloxacin (CIP), and vancomycin (VAN) (Table S8). Combining the antibiotic resistance phenotypes and resistance genes, we found that the strains carrying multiple antibiotic resistance genes also in general had the multidrug resistance phenotype. There was a pronounced correlation between antibiotic resistance phenotypes and some specific CGs. All of the isolates of CG7 exhibited consistent resistance to MEM, CLI, TET, DOX, CTX, GEN, CIP, ERY, and PEN. All of the isolates of CG8 exhibited consistent resistance to MEM, CLI, TET, DOX, CTX, GEN, CIP, ERY, and PEN. All of the isolates of CG12 exhibited consistent resistance to MEM, CLI, CTX, CIP, ERY, and PEN. All of the isolates of CG13 exhibited consistent resistance to CIP. All of the isolates of CG16 exhibited consistent resistance to MEM, CLI, CTX, CIP, ERY, and PEN. Thirty-four isolates of CG4 (81.0%) exhibited consistent resistance to MEM, CLI, RIF, CTX, GEN, CIP, ERY, and PEN.

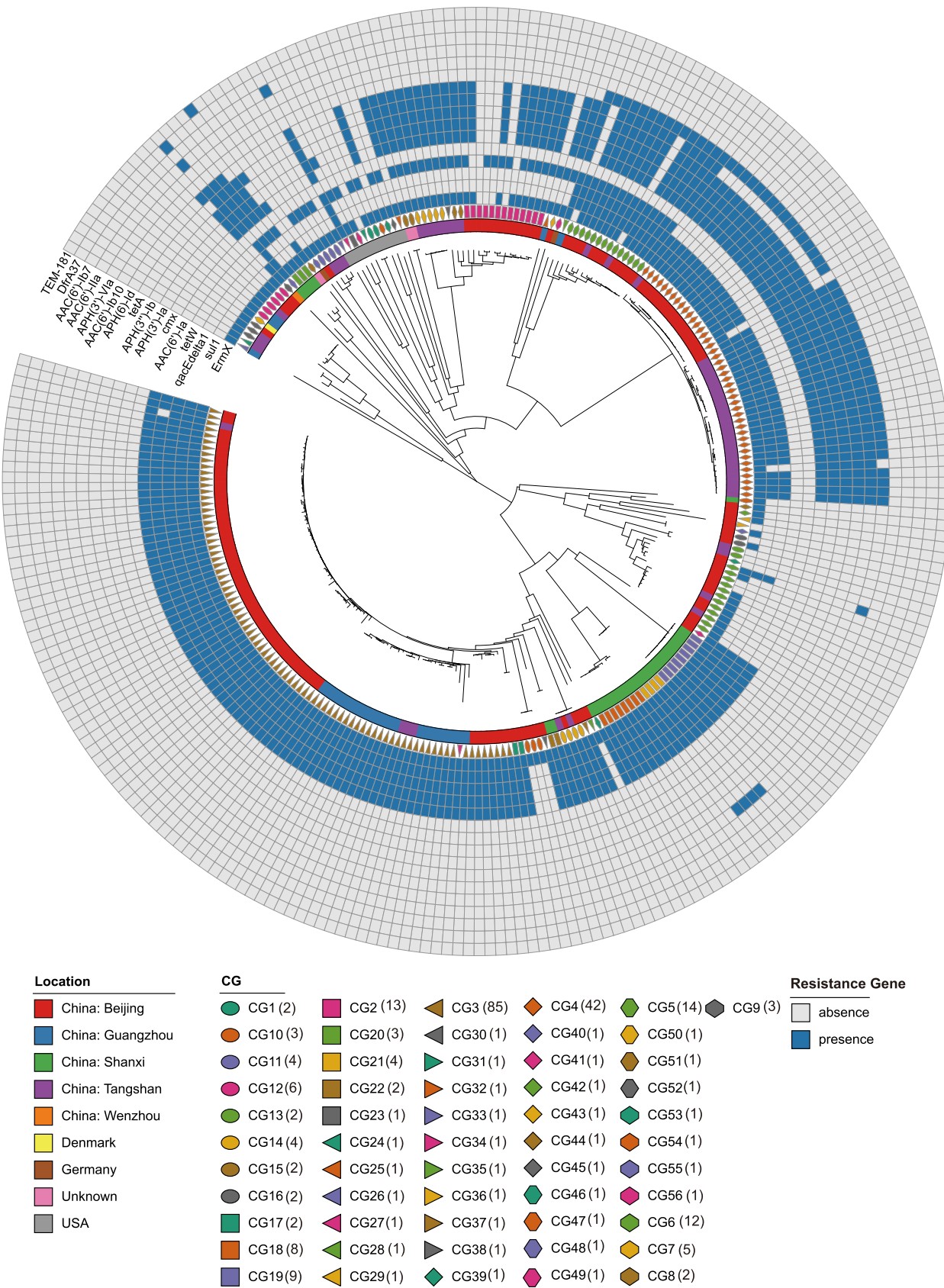

**FIG 6** Neighbor-joining tree of the 263 *C. striatum* genomes as determined on the basis of cgMLST allelic profiles, showing the association between countries, CGs, and resistance features.

Most isolates of CG3 had similar antibiotic resistance phenotypes. Thirty-seven isolates of CG3 (31.8%) exhibited consistent resistance to MEM, CLI, RIF, CTX, GEN, CIP, ERY, and PEN. Thirty-seven isolates of CG3 (31.8%) exhibited consistent resistance to MEM, CLI, TET, CTX, CIP, ERY, and PEN.

**Presence and diversity of virulence genes among *C. striatum* strains.** In order to investigate the potential association between the presence/absence of virulence genes and CGs, we searched for potential virulence genes in the Virulence Factor Database (VFDB). A total of three virulence genes were identified in all of the strains (Fig. 7). It is noteworthy that the *spaE*, *srtB*, and *srtC* genes are involved in adhesion. We found a high proportion of strains (47.5% to 46.0%) harboring the *spaE*, *srtB*, and *srtC* genes. Some difference in the carriage of virulence genes was found among different CGs. All isolates of 25 CGs (CG2, CG4, CG5, CG7, CG8, CG11, CG14, CG15, CG18, CG19, CG20, CG23, CG24, CG25, CG26, CG28, CG29, CG30, CG32, CG43, CG48, CG49, CG51, CG52, and CG55) harbored all three virulence genes. In contrast, CG3, CG6, CG9, CG10, CG12, CG13, CG16, CG17, CG21, CG22, CG27, CG31, CG33, CG34, CG35, CG36, CG37, CG38, CG39, CG40, CG41, CG42, CG44, CG50, CG53, CG54, and CG56 were entirely devoid of virulence genes.

## DISCUSSION

The characteristics of multidrug resistance and horizontal transmission of *C. striatum* isolates suggest that it is likely to be an emerging nosocomial pathogen (43). Until now, however, a consensus approach to characterize and compare *C. striatum* isolates has been lacking, thus limiting our understanding of the biology and epidemiology of strains of this potential pathogen and hindering our ability to establish appropriate control and prevention measures. Several molecular typing methods, such as DNA fingerprinting, matrix-assisted laser desorption ionization–time of flight mass spectrometry (MALDI-TOF MS), PFGE, and MLST, have been used to demonstrate an epidemiological connection between the isolates in *C. striatum* outbreaks in previous studies (11–14, 19–21). However, these typing methods provided limited resolution. In this study, we proposed a cgMLST scheme for *C. striatum*, a methodology that combines whole-genome sequencing data with traditional MLST principles. The proposed cgMLST scheme using 1,795 core genes offers an enhanced level of resolution, which improves the ability to distinguish among *C. striatum* isolates. This study sets the stage for a comprehensive understanding of the biodiversity of *C. striatum* and for the epidemiological surveillance of medically important *C. striatum* strains.

This is the first study with a large population size to characterize the population structure and evolution relationship of *C. striatum*. The isolates in the present study were divided into 56 CGs. Two CGs were the most abundant: CG3 and CG4, occurring in 85 and 42 strains, respectively. The analysis of virulence genes and drug resistance genes indicates that CG4, CG5, CG26, CG28, and CG55 represent hypervirulent and multidrug-resistant CGs of *C. striatum*. Of concern is that CG4, CG5, CG28, and CG55 currently only appear in areas of China. Particular attention should be paid to nosocomial infections caused by CG4, CG5, CG28, and CG55 isolates. To the best of our knowledge, this is also the first study to investigate the CGs corresponding to highly virulent and multidrug-resistant *C. striatum* isolates based on a cgMLST analysis.

Based on the present sample of *C. striatum* genomes, the geographic distributions of some CGs showed certain differences: every country and even every city in China had its own unique CGs. Although no CG was shared by different countries, some CGs were shared by different cities in China (Beijing, Tangshan, and Guangdong). The reason for Beijing and Tangshan sharing the most CGs was probably their close geographic distance. The straight-line distance between Beijing and Tangshan is only 155 km. We speculated that the transmission of CGs of *C. striatum* was possibly related to the flow of carriers between different regions. Due to the vast majority of the 263 strains used to establish and validate the cgMLST scheme being from China (246 [93.54%]), our future studies will collect more *C. striatum* isolates from foreign countries

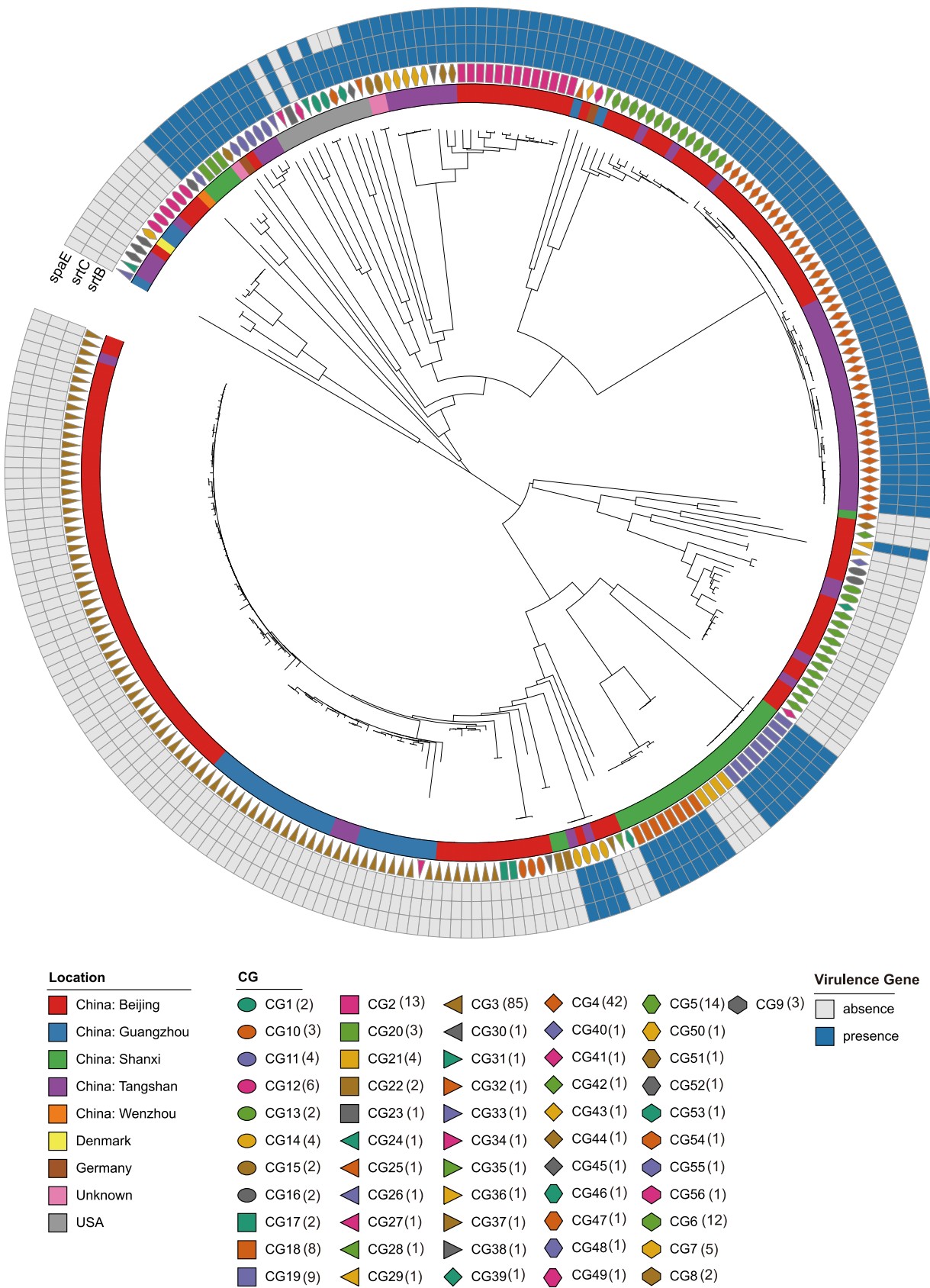

**FIG 7** Neighbor-joining tree of the 263 *C. striatum* genomes as determined on the basis of cgMLST allelic profiles, showing the association between countries, CGs, and virulence features.

to explore whether transmission of *C. striatum* CGs occurred over large geographic distances and to validate of the applicability of our cgMLST scheme for investigating outbreaks of *C. striatum* strains in foreign countries.

We analyzed 30 clinical isolates from 26 patients collected in a hospital in Shanxi, China, between March and April 2021 to validate the usefulness of our cgMLST scheme in investigating hospital transmission. These *C. striatum* isolates were largely isolated from respiratory samples, which is consistent with most previous studies (13–15). CG analysis based on a threshold of 104 allelic differences revealed that CG18 and CG19 may be the main prevalent CGs in this hospital. Hospital transmission analysis based on a threshold of 7 allelic differences found 4 clusters with 2 or more isolates. Since isolate 3 and isolate 6 were isolated from the same patient, a total of 3 potential hospital transmission events were inferred. The strains in cluster 31 and cluster 32 were isolated from different units and different buildings, highly suggestive of *C. striatum* transmission within and between hospital units, even buildings. Earlier genotype studies have confirmed patient-to-patient transmission of *C. striatum* via caretakers in hospitals (11). A recent study by Wang et al. found potential within-hospital and between-hospital transmission of *C. striatum* (34).

Isolate 14 and isolate 19, from patients with community-acquired infections, and epidemiologically unrelated isolate 72 formed three separate clonotypes and transmission clusters. In contrast, we found that *C. striatum* isolates isolated from the same patient had the same CG patterns and belonged to the same transmission cluster, such as isolate 3 and isolate 6 (isolated from patient 3), isolate 20 and isolate 23 (isolated from patient 16), and isolate 29 and isolate 33 (isolated from patient 23). Therefore, 1,795-cgMLST has the potential to be used as an accurate and precise epidemiological typing tool for hospital transmission investigation.

In our validation step, infections by closely related isolates occurred in different wards or even different buildings. This is consistent with findings of Wang et al. (18). The transmission occurred probably between patients, between health workers and patients, or through the environment or fomites. A hospital outbreak is never a coincidence but rather reflects a failure of the infection control measures (14). We believe that rapid confirmation of the spread of a single clone/type of a *C. striatum* isolate within or across several hospital units by an available cgMLST scheme may contribute to improving the management of an outbreak. This study is the first to document the usefulness of the cgMLST scheme as a typing tool for investigating nosocomial outbreaks of *C. striatum*.

Antimicrobial multidrug-resistant *C. striatum* strains have been increasingly related to nosocomial outbreaks worldwide. Multidrug-resistant strains isolated both in the hospital environment and in the community have become a public health concern among epidemiologists and the entire medical community (44). In the present study, 79.1% of isolates (208/263) carried more than 2 antibiotic resistance genes. Combining the antibiotic resistance phenotype and resistance genes, we found that the strains carrying multiple antibiotic resistance genes also in general had the multidrug resistance phenotype. Growing numbers of nosocomial infections caused by multidrug-resistant *C. striatum* clones have been documented in several countries, including Italy, Spain, the Netherlands, the United States, China, and Japan (9, 19, 45, 46). Further studies are needed to elucidate the mechanisms of resistance of *C. striatum* strains to antimicrobial agents.

Opportunistic pathogens may have an array of virulence factors that contribute to their ability to survive within host tissues and confer resistance to host immune defenses and clearance mechanisms and antimicrobial killing. In total, we annotated three virulence-associated genes among 263 *C. striatum* strains. Of note, all of these virulence-associated genes were associated with adhesion. Host attachment is the first step, and an essential step, in the bacterial infection process (47). A study conducted by Alibi et al. showed that *C. striatum* strains exhibited good adhesion to inert surfaces and human epithelial cells (48). In Gram-positive bacteria, cell surface structures, including pili, play an important role in the colonization of host tissues, which is thought

to be a critical step during infection (49). In addition, microbial adhesion is considered the first step for biofilm formation (50). Previous studies based on wet experiments have validated the biofilm-forming ability of *C. striatum* (21, 51, 52). The ability of some *C. striatum* clones to produce biofilm on different types of surfaces may contribute to its pathogenicity.

With the continual improvement of next-generation sequencing techniques and the decrease of the cost of strain genome sequencing (renminbi [RMB], 100 to 300 yuan/genome), whole-genome sequencing is expected to become an identification tool used in clinical and epidemiological studies. It is of vital importance to develop a rapid typing scheme for genome sequence data that is capable of investigating the relationship of a novel isolate with known strains. In this study, we created a cgMLST scheme for *C. striatum* for the first time. We have made our scheme and associated files available at GitHub (https://github.com/Natasha22222222/cgMLST-Corynebacterium-triatum) aiming to foment discussions and possibly help in establishing a cgMLST consensus for *C. striatum*. Gene target lists and selected loci are available at this location. In the GitHub link, we also present a step-by-step explanation of how new genome sequences may be analyzed to make this method be applied to other outbreak situations. The whole analysis is expected to be completed within 24 h when the sample size is fewer than 500 genomes. The remaining challenge is to establish an Internet-based nomenclature server, such as EnteroBase (http://enterobase.warwick.ac.uk/), to facilitate universal global nomenclature and provide an outbreak investigation tool for any user.

**Conclusions.** In this study, through cgMLST analysis of 1,795 highly conserved genes, we performed CG typing of 263 *C. striatum* isolates and investigated potential hospital transmission events. We identified five potential hypervirulent and multidrug-resistant CGs of *C. striatum* based on genome-wide analysis. Taken together, our results indicate that our proposed cgMLST scheme may be adequate for epidemiology and surveillance approaches and hospital transmission investigations associated with *C. striatum*.

## MATERIALS AND METHODS

**Ethics approval.** The Research Ethics Committee of the Chinese Center for Disease Control and Prevention (no. ICDC-2018005) approved this study. All of the procedures in this study were performed in accordance with the Declaration of Helsinki.

**Isolate selection for genome sequencing.** Genomic sequences available as of 29 August 2021 for *C. striatum* were downloaded from the NCBI Genome Database (http://www.ncbi.nlm.nih.gov/assembly/). Two hundred seventy-five genomes were downloaded in total, including seven complete genomes and 263 whole-genome shotgun sequences available as scaffolds or contigs. Among the 263 draft genomes, 3 were discarded because of more than 5% contamination, and 1 was discarded because of less than 85% completeness. Thus, a total of 271 publicly available genome sequences were used in this study, and they are described in Table S1.

**Sequenced strains and analyzed whole-genome sequences.** A clinical sample was obtained from hospitalized patients growing exclusively *C. striatum* colonies, according to the definition of an outbreak case described by Verroken et al. (14). A total of 30 *C. striatum* isolates were isolated from 26 patients from a hospital in Shanxi, China, between 24 March and 23 April 2021. One epidemiologically unrelated outgroup strain was collected from a hospital in Wenzhou, China, in January 2019. Genomic DNA was isolated and purified using the Wizard genomic DNA purification kit (Promega, WI, USA). Whole-genome sequencing was carried out with the Illumina HiSeq X 10 sequencing platform (150-bp paired-end reading). Trimmomatic version 0.39 (53) was used to remove adapter sequences and low-quality sequences. Assembly was performed using SPAdes v3.11.1 (54). The CheckM tool was used to assess genome completeness and contamination (55).

**Core genome multilocus sequence typing scheme.** A training file was created in Prodigal v2.6.3 (56) with *C. striatum* WP1a (RefSeq assembly accession number GCF_004138065.1), and this file was used in the next steps of the study. All 271 *C. striatum* strains were assessed with the CreateSchema module in the chewBBACA (57) suite, during which time each genome was annotated individually. In brief, in the first step, the algorithm defined coding sequences for every genome, performed comparisons of them in a pairwise fashion, and generated a single FASTA file containing the selected coding sequences. In the second step of whole-genome MLST (wgMLST) creation, the allele-calling operation (AlleleCall) recognized and excluded loci that were possible paralogs. Then, the remaining list of loci was used to define the cgMLST scheme. The quality test of the wgMLST scheme was conducted using the TestGenomeQuality module. Candidate loci for the cgMLST scheme were extracted from the wgMLST scheme using the ExtractCgMLST operation with a coding sequence presence threshold of 95%. cgMLST schemata are defined as the set of loci that are present in all strains under analysis, but defining a smaller loci presence threshold, such as 95%, might be necessary to include very frequent genes or ubiquitous genes that are not present in some strains due to sequencing/assembly limitations. We selected

this threshold value to account for loci that may not be identified in incomplete sequences due to sequencing coverage and assembly problems. In this stage, we chose the threshold, a value of 15, that limited the loss of the loci in the genomes. Beginning at a threshold of 15, the number of the loss of the loci in the genomes tended to plateau (Fig. S1). In accordance with a study by de Sales et al. (58), in the validation step, we kept candidate loci common to 99% of the isolates and chose the threshold (i.e., 15) that limited the loss of the loci in the genomes. Beginning at this threshold, the number of the loss of the loci in the genomes tended to plateau (Fig. S2). This cgMLST scheme is publicly available at https://github.com/Natasha22222222/cgMLST-Corynebacterium-triatum.

**Graphical representation of cgMLST results.** Minimum spanning trees (MST) based on allelic profiles were created using GrapeTree (version 2.1) with the MSTreeV2 method (59). Furthermore, phylogenetic neighbor-joining trees were calculated using the neighbor-joining algorithm (StandardNJ) implemented in GrapeTree (version 2.1). Trees were visualized and annotated using the iTOL online platform (https://itol.embl.de/).

**Virulome and resistome analysis.** Antimicrobial resistance genes and virulence genes were screened in each *C. striatum* genome using the ABRicate pipeline (https://github.com/tseemann/abricate) on the Comprehensive Antibiotic Resistance Database (CARD) (https://card.mcmaster.ca/) and Virulence Factor Database (VFDB) (http://www.mgc.ac.cn/VFs/). Only query genes with an identity higher than 90% and a coverage higher than 90% were considered potential antimicrobial resistance genes or virulence genes.

**Comparison between core genome MLST and SNP typing.** Phylogenetic analysis of core genome single-nucleotide polymorphisms (SNPs) was performed using nucleotide sequences of all 1,795 cgMLST genes. For each core gene, multiple-sequence alignment was performed with MAFFT (60), and then the resulting gene alignments were concatenated in a supermatrix core genome alignment. SNP sites were extracted from aligned sequences using snp-sites (61). A phylogenetic tree was inferred using the maximum likelihood (ML) method in FastTree software (62). Topological concordance between the SNP-based ML tree and cgMLST-based neighbor-joining tree was visualized using the tanglegram algorithm (63) in Dendroscope software (64).

**MLST.** Since the most of genomes in our cgMLST scheme were downloaded from the NCBI database, the sequence of the housekeeping gene could not be obtained by PCR amplification. The sequences of 6 housekeeping genes (*gyrA*, *gyrB*, *hsp65*, *rpoB*, *secA1*, and *sodA*) were extracted from 263 *C. striatum* genomes using usearch -search_pcr2 (http://www.drive5.com/usearch/). The primer sequences are listed in Table S2. Alleles of each housekeeping gene were randomly numbered, and the six studied loci were then used to determine the allelic profiles. A specific sequence type (ST) was determined for each isolate based on the allelic profile of six housekeeping genes.

**Data availability.** The genomic sequencing data set was deposited in the NCBI Sequence Read Archive (SRA) under BioProject accession number PRJNA825744.

## SUPPLEMENTAL MATERIAL

Supplemental material is available online only.

**SUPPLEMENTAL FILE 1**, PDF file, 1.4 MB.

**SUPPLEMENTAL FILE 2**, XLSX file, 0.1 MB.

## ACKNOWLEDGMENTS

We thank Xuebing Wang for sharing the antibiotic resistance phenotypes and PFGE data with us.

We declare no competing interests.

This work was supported by grants from the National Key R&D Program of China (grant no. 2019YFC1200700), the National Natural Science Foundation of China (grant no. 82073624), and the National Science and Technology Major Project (grant no. 2018ZX10201001).

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
