## [Reviewer comments · Microbiology Spectrum]

Microbiology Spectrum

Epidemiological investigation of hospital transmission of *Corynebacterium striatum* Infection by Core Genome Multilocus Sequence Typing approach

Yutong Kang, Shenglin Chen, Leihao Tian, Xiaoli Du, Zhenpeng Li, Zhizhou Tan, Haijian Zhou, Jia Huang, Beijia Zheng, Jiaxin Zhong, Xueli Ma, fang li, Yao Jiang, Yu Wang, Meiqin Zheng, and Zhenjun Li

Corresponding Author(s): Yutong Kang, Wenzhou Medical University

Review Timeline:

Submission Date:	April 24, 2022
Editorial Decision:	August 17, 2022
Revision Received:	November 4, 2022
Accepted:	November 17, 2022

Editor: Daria Van Tyne

Reviewer(s): The reviewers have opted to remain anonymous.

Transaction Report:

DOI: <https://doi.org/10.1128/spectrum.01490-22>

August 17, 2022

Dr. Yutong Kang
Wenzhou Medical University
Wenzhou
China

Re: Spectrum01490-22 (Epidemiological investigation of a nosocomial outbreak of *Corynebacterium striatum* Infection by Core Genome Multilocus Sequence Typing approach)

Dear Dr. Yutong Kang:

Thank you for submitting your manuscript to Microbiology Spectrum. I apologize for the long delay in getting your manuscript reviewed, it was unusually difficult to find reviewers. Your study has been reviewed by two experts, and I would now like you to revise your manuscript in line with their feedback. Both reviewers recognize the value in your work, but both also identified several areas that are in need of additional attention. Please be sure to address all of the reviewer comments in your revised submission. Additionally, while you have utilized a large number of publicly available genomes, the genomes you have newly sequenced in this study must also be made publicly available in order to comply with the journal's data availability policy. Please deposit your genomes into an appropriate public database (like NCBI) and include a "Data availability" section at the end of the Methods that lists accession information for the newly sequenced genomes.

Link Not Available

Sincerely,

Daria Van Tyne

Journals Department
Reviewer comments:

Reviewer #1 (Comments for the Author):

Kang et al. perform an epidemiological investigation of a nosocomial *C. striatum* outbreak in China using Core Genome Multilocus Sequence Typing. The study is well designed and the manuscript is well written. A few comments to improve the

submission are below.

1. In the Introduction, additional information and background regarding *C. striatum* is necessary to establish the importance of this study. Additional details should be provided regarding environmental reservoirs, transmission and range of patient disease presentation and prognosis in order to establish the importance of this work and study. I would recommend expanding on these details and moving the study summary (last paragraph of Intro) to the conclusions.
2. Please provide more details about the hospital settings.
3. An important point in this study is that the vast majority of the 271 strains used to establish the cgMLST scheme were from China. Beyond speculation about transmission of *C. striatum* CGs, additional discussion is needed as to how this biases results, considering the outbreak strains evaluated were also isolated from patients presenting to Chinese hospitals, and potential limitations of this method in this regard given association between country and CG. Do the authors believe this evaluation reflects the true biodiversity of *C. striatum* isolates? Although this is touched on in the discussion, the applicability of this assessment and implications remain unclear.
4. More practically, how can cgMLST be applied in hospital settings to identify outbreaks and direct treatment? A summary of time to result, analysis, price, throughput, etc. would be helpful, including whether this method can be applied to other outbreak situations, further increasing the value of this methodology in a clinical setting.

Minor

- define "cg" in "cgMLST" as well as basic principle of method at first mention (ln 87)
- did patients provide informed consent for WGS and data analysis?
- Why were the selected thresholds for results and analysis selected? Some rationale for these levels would be helpful in the methods section (CG definition is well defined but other thresholds are not)
- unclear what "file" refers to in Table S2
- In 203 and 227, change "correspondence" to "correlation"
- please check presentation of figure S2 (currently cut-off)
- Where are references to figure S1, S2 in text?

Reviewer #2 (Comments for the Author):

Kang et al provide a well written description of a genomic analysis of *Corynebacterium striatum*, a bacterial species not well studied, yet capable of human infection and potential patient-to-patient spread in hospital settings. They devised a cgMLST scheme for this species, which could have utility in discriminating *C. striatum* clones in future epidemics. Using 271 publicly available *C. striatum* genomes, they produced a cgMLST scheme with 1,776 conserved genes, which produced 56 clonal groups (CG) among this dataset. To attempt to demonstrate the potential of this scheme, they sequenced 30 genomes isolated from 26 patients at a single hospital over a 1-month period and characterized the CGs represented. They also explored resistance and virulence genes among the larger dataset to test the power of their scheme to delineate groups with different resistance and/or virulence genotypes. While the authors have done an adequate job in creating their cgMLST scheme, the analyses demonstrating its utility are not strong enough to support the claims that are made. Some important changes needed to improve the manuscript include a better description of what the authors refer to as an "outbreak" and how the cgMLST scheme helps to delineate one, comparisons of cgMLST to other methods of typing and transmission detection, and better connections to pathogenicity potential of specific CGs that are grounded in what has previously been described for this species.

Major comments:

The authors have sufficiently devised a cgMLST scheme. However, they have not adequately demonstrated its utility in describing an outbreak of virulent/resistant strains based on the analyses they have conducted here. Specifically, they refer to an "outbreak" that resulted in the 30 isolates from one hospital. However, the authors need to better define what they mean by outbreak. This term generally refers to a notable increase in cases over a specified period of time, which is alluded to, but is not well described. What was the prevalence before and after this timeframe? Do the 26 cases described represent all cases from that ~1 month period? How were these samples obtained? Were the patients already hospitalized before being infected? Were they from diseased individuals with an infection caused by *C. striatum*? Colonized individuals? This is particularly important because in the follow-up CG analyses, there appears to be a great deal of diversity among these 26 cases, which conflicts with the idea that there is a notable increase in transmission. Based on this, it is questionable to conclude that cgMLST can distinguish the 30 "outbreak-associated" isolates from the two isolates chosen from a different location and time as not outbreak-associated. How do we know the outbreak strains, as defined here, are not just the endemic population?

Related to the above comment, there are significant issues with Figure 5a. First, the scale on the x axis does not represent a continuous flow of time as the dates are not evenly spaced. Therefore, attempting to describe changes over time is not accurate. Further, it is unclear why cumulative cases are being plotted here. When characterizing outbreaks/epidemics, rates of change are far more valuable. Finally, all 30 isolates are plotted on this graph when there are only 26 patients. Unless multiple isolates from the same patient represent multiple episodes, this presents bias. In describing this panel on lines 218-219, the authors refer to the slow increase suggests nosocomial infections and transmission between people. This is not clear and should

either be removed or explained in greater detail to support such a significant statement.

While the CGs groups defined here might have utility for future studies, the authors do not do enough to compare their results to what has been previously described in this species to verify just how discriminatory their method is. At a minimum, STs should be identified for all isolates. Including some measure of SNP distance, k-mer distance, or average nucleotide identity would help also. 56 CGs sounds like a lot of diversity. However, there is no context, either to previous studies in *C. striatum* or other species. It does look highly probable that there is transmission, particularly for CGs 16, 17, and 19. In many of these cases, the dates of similar isolates are quite close and often from the same units. If the ideal use of cgMLST is to capture cases of transmission that produce outbreaks, it would improve the manuscript to study these cases further and compare the cgMLST scheme to other methods traditionally used for transmission detection, such as SNP distances. Is it possible that a single CG represents a transmission cluster? That would be quite valuable. Another reason to place these isolates in the context of previous studies is to better understand their pathogenic traits. Presumably, certain STs or PFGE pulsotypes have been associated with particular outcomes or phenotypes (like ref. 8). The analyses presented by the authors do identify resistance and virulence genes, with some references to previous studies; however, there should be more direct connections to studies that have done wet-lab work.

There is a general lack of citations for some claims in the introduction, e.g. lines 61-65, 81-88

There should be more quantitative metrics of genome quality. What were the contamination and completeness %, number of contigs, and N50 values for each genome? Fragmented genomes will certainly affect the ability to annotate the genes appropriately, which would impact what is identified as core. Using genomes with N50s below, say, 25kb or over 300 contigs could have pretty significant impacts.

The identity and coverage thresholds for abricate are too low. Values in the range of 90-95% are more in line with current research when it comes to individual genes.

Some of the claims regarding geographic barriers of transmission should be tempered and heavily caveated, e.g. lines 210-213. It is notable that there is no overlap between countries. However, the total number of isolates from outside of China is incredibly small and certainly does not represent the diversity that exists. If the authors want to try and better support this claim, they could try to look at the distribution of CGs within China to see if there is a lack of mixing of CGs in regions in the country.

Minor comments:

Line 32: please change to "known diversity of this species"

Lines 31, 38-39, 94, 347-349, etc: As mentioned previously, "outbreak" is not an appropriate word here until better defined.

Line 42: Since there is no phenotypic or clinical outcome data, these CGs are "potentially hypervirulent and multidrug-resistant"

Lines 62 and 65: please provide examples of the "specific circumstances" and "range of diseases" with citations.

Lines 87-88: please cite cgMLST here.

Line 91: do the authors mean "combining?"

Lines 228-233: There are many cases when it seems "isolate" should be singular, correct? It would help to better describe that CS refers to an isolate. Or better yet, the authors could remove CS altogether and refer to isolates with just a number. As an example: "With the exception of isolate 34 from patient 23..."

Lines 341-343: It is important to note that *C. striatum* isolates could also be re-introduced into these hospital units from environmental sources as well.

Lines 354-359: These are bold claims that have not been supported by the evidence presented. These should either be removed or heavily caveated by the fact that the authors did not find anything to support this.

Staff Comments:

Preparing Revision Guidelines

To submit your modified manuscript, log onto the eJP submission site at <https://spectrum.msubmit.net/cgi-bin/main.plex>. Go to Author Tasks and click the appropriate manuscript title to begin the revision process. The information that you entered when you

first submitted the paper will be displayed. Please update the information as necessary. Here are a few examples of required updates that authors must address:

Please return the manuscript within 60 days; if you cannot complete the modification within this time period, please contact me. If you do not wish to modify the manuscript and prefer to submit it to another journal, please notify me of your decision immediately so that the manuscript may be formally withdrawn from consideration by Microbiology Spectrum.

Dear Editor and reviewer

We would like to thank you for all the positive comments on our manuscript (No. Spectrum01490-22) entitled “Epidemiological investigation of a nosocomial outbreak of *Corynebacterium striatum* Infection by Core Genome Multilocus Sequence Typing approach”. We are truly grateful for the reviewer’s critical comments and thoughtful suggestions on our manuscript, which have enabled us to improve our work. All of the comments and suggestions have been seriously considered, and significant improvements have been made in the revised manuscript. The accession information for the newly sequenced genomes has been listed in the "Data availability" section at the end of the Methods. The corresponding revisions in the body of manuscript are marked in red.

We hope the new manuscript will meet your journal’s standards. If you have any questions, please do not hesitate to contact me.

Sincerely yours,

Zhenjun Li on behalf of all the authors

Reviewer #1

Major comments:

1. In the Introduction, additional information and background regarding *C. striatum* is necessary to establish the importance of this study. Additional details should be provided regarding environmental reservoirs, transmission and range of patient disease presentation and prognosis in order to establish the importance of this work and study. I would recommend expanding on these details and moving the study summary (last paragraph of Intro) to the conclusions.

Response: Thank you for your advice. We have rewritten and moved the last paragraph of introduction to the conclusions (lines 82 to 92). We expanding on details of environmental reservoirs, transmission and range of patient disease presentation and prognosis in first paragraph of introduction (lines 552 to 558).

2. Please provide more details about the hospital settings.

Response: Thank you for your advice. We have provided more details about the hospital settings in lines 61 to 74 (lines 301 to 316 and lines 339 to 341).

3. An important point in this study is that the vast majority of the 271 strains used to establish the cgMLST scheme were from China. Beyond speculation about transmission of *C. striatum* CGs, additional discussion is needed as to how this biases results, considering the outbreak strains evaluated were also isolated from patients presenting to Chinese hospitals, and potential limitations of this method in this regard given association between country and CG. Do the authors believe this evaluation reflects the true biodiversity of *C. striatum* isolates? Although this is touched on in the discussion, the applicability of this assessment and implications remain unclear.

Response: Thank you for your advice. We have added descriptions of these potential limitations in the discussion section. We also have looked at the distribution of CGs within China to see the mixing of CGs in regions in the country (lines 241 to 255 and lines 463 to 475).

4. More practically, how can cgMLST be applied in hospital settings to identify outbreaks and direct treatment? A summary of time to result, analysis, price, throughput, etc. would be helpful, including whether this method can be applied to other outbreak situations, further increasing the value of this methodology in a clinical setting.

Response: Thank you for your advice. In the revised manuscript, we defined an outbreak threshold based on seven allelic differences, which is capable of identifying hospital transmission-related isolates. We have made our scheme and associated files available on GitHub (<https://github.com/Natasha22222222/cgMLST-Corynebacterium-striatum>). In the GitHub link, we also present a step-by-step explanation of how new genome sequences may be analyzed to make this method be applied to other outbreak situations. We summarize the time to result, analysis, price, throughput in the Discussion section. Please find below responses to all your comments.

In lines 534 to 550, “With the continual improvement of next-generation sequencing techniques and the decrease of the cost of strains genome sequencing (RMB: 100-300 yuan/genome), whole-genome sequencing is expected to become an identification tool used in clinical and epidemiological studies. It is of vital importance to develop a rapid typing scheme for genome sequence data that is capable of investigating the relationship of a novel isolate with known strains. Here, we created a cgMLST scheme for *C. striatum* for the first time. We have made our scheme and associated files available at GitHub (<https://github.com/Natasha22222222/cgMLST-Corynebacterium-striatum>) aiming to foment discussions and possibly help in establishing a cgMLST consensus for *C. striatum*. Gene target lists and selected loci are available at this location. In the GitHub link, we also present a step-by-step explanation of how new genome sequences may be analyzed to make this method be applied to other outbreak situations. The whole analysis is expected to be completed within

24 hours when the sample size of fewer than 500 genomes. The remaining challenge is to establish an Internet-based nomenclature server, such as EnteroBase (<http://enterobase.warwick.ac.uk/>), to facilitate universal global nomenclature and provide an outbreak investigation tool for any user.”

Minor

-define "cg" in "cgMLST" as well as basic principle of method at first mention (ln 87)

Response: Thank you for your advice. We have defined "cg" in line 48. The basic principle of "cgMLST" were described in lines 112 to 115.

-did patients provide informed consent for WGS and data analysis?

Response: Thank you for your advice. Yes, all patients agreed to participate and signed informed consent forms.

-Why were the selected thresholds for results and analysis selected? Some rationale for these levels would be helpful in the methods section (CG definition is well defined but other thresholds are not)

Response: Thank you for your advice. We have added some rationale for the selected thresholds for results and analysis in lines 156 to 170.

-unclear what "file" refers to in Table S2.

Response: Thank you for your reminder. We have modified Table S2 as follow: first column is GeneBank accession no., second column is strain names, third column is CG types.

-In 203 and 227, change "correspondence" to "correlation"

Response: Thank you for your advice. We have changed "correspondence" to "correlation" and rewritten this sentence as follow: There was a pronounced correlation between locations and some specific CGs (lines 245 to 246).

-please check presentation of figure S2 (currently cut-off)

Response: Thank you for your reminder. We have added the description of cut-off in the figure S2 in lines 168 to 170.

-Where are references to figure S1, S2 in text?

Response: Thank you for your reminder. Response: Thank you for your reminder. We have added the references to figure S1, S2 in text (lines 165 to 170).

Reviewer #2 (Comments for the Author):

Major comments:

The authors have sufficiently devised a cgMLST scheme. However, they have not adequately demonstrated its utility in describing an outbreak of virulent/resistant strains based on the analyses they have conducted here. Specifically, they refer to an "outbreak" that resulted in the 30 isolates from one hospital. However, the authors need to better define what they mean by outbreak. This term generally refers to a notable increase in cases over a specified period of time, which is alluded to, but is not well described. What was the prevalence before and after this timeframe? Do the 26 cases described represent all cases from that ~1 month period? How were these samples obtained? Were the patients already hospitalized before being infected? Were they from diseased individuals with an infection caused by *C. striatum*? Colonized individuals? This is particularly important because in the follow-up CG analyses, there appears to be a great deal of diversity among these 26 cases, which conflicts with the idea that there is a notable increase in transmission. Based on this, it is questionable to conclude that cgMLST can distinguish the 30 "outbreak-associated" isolates from the two isolates chosen from a different location and time as not outbreak-associated. How do we know the outbreak strains, as defined here, are not just the endemic population?

Response: We gratefully appreciate for your valuable comment. We are very sorry for our improper understanding of the outbreak. In-hospital prevalence of *C. striatum* infection was relatively stable each month. About 30-40 strains can be isolated every

month. In our present study, we only collected all strains isolated from March 24 to April 23, 2021 for whole-genome sequencing to validate our cgMLST scheme. The results of Wang et al. also demonstrated that *C. striatum* has spread in parallel across China, causing persistent and extensive transmissions within hospitals (1). Therefore, in our revised manuscript, we defined an outbreak threshold based on seven allelic differences that is capable of identifying strains from the same outbreak and closely related isolates, to investigate the nosocomial spread of *C. striatum*.

A total of 30 *C. striatum* isolates were obtained from 26 patients. Two of the 26 patients had been hospitalized for lung infections. The remaining 24 patients had been hospitalized for some underlying condition and contracted *C. striatum* during their stay. Two or more times cultures were positive in 3 patients. Among them, 7 isolates were originated from tracheal secretion, 22 isolates from sputum, 1 isolate from catheter. Since *C. striatum* was as a pure culture growth from these clinical specimens and the patients had symptoms of infection, we believe that *C. striatum* isolated from all patients was the pathogen of infection, not the colonizer (lines 286 to 300).

Ref (1): Wang X, Zhou H, Du P, Lan R, Chen D, Dong A, Lin X, Qiu X, Xu S, Ji X.

2021. Genomic epidemiology of *Corynebacterium striatum* from three regions of China: an emerging national nosocomial epidemic. *J Hosp Infect* 110:67–75.

Related to the above comment, there are significant issues with Figure 5a. First, the scale on the x axis does not represent a continuous flow of time as the dates are not evenly spaced. Therefore, attempting to describe changes over time is not accurate. Further, it is unclear why cumulative cases are being plotted here. When characterizing outbreaks/epidemics, rates of change are far more valuable. Finally, all 30 isolates are plotted on this graph when there are only 26 patients. Unless multiple isolates from the same patient represent multiple episodes, this presents bias. In describing this panel on lines 218-219, the authors refer to the slow increase suggests nosocomial infections and transmission between people. This is not clear and should either be removed or explained in greater detail to support such a significant

statement.

Response: We gratefully appreciate for your valuable comment. We are very sorry for our wrong Figure 5a. In the revised manuscript, we have substitute figure 5a into a grid plot to exhibited for the isolation time, patients, and potential transmission of *C. striatum* isolates. Corresponding text has been also revised.

While the CGs groups defined here might have utility for future studies, the authors do not do enough to compare their results to what has been previously described in this species to verify just how discriminatory their method is. At a minimum, STs should be identified for all isolates. Including some measure of SNP distance, k-mer distance, or average nucleotide identity would help also. 56 CGs sounds like a lot of diversity. However, there is no context, either to previous studies in *C. striatum* or other species. It does look highly probable that there is transmission, particularly for CGs 16, 17, and 19. In many of these cases, the dates of similar isolates are quite close and often from the same units. If the ideal use of cgMLST is to capture cases of transmission that produce outbreaks, it would improve the manuscript to study these cases further and compare the cgMLST scheme to other methods traditionally used for transmission detection, such as SNP distances. Is it possible that a single CG represents a transmission cluster? That would be quite valuable. Another reason to place these isolates in the context of previous studies is to better understand their pathogenic traits. Presumably, certain STs or PFGE pulsotypes have been associated with particular outcomes or phenotypes (like ref. 8). The analyses presented by the authors do identify resistance and virulence genes, with some references to previous studies; however, there should be more direct connections to studies that have done wet-lab work.

Response: We gratefully appreciate for your valuable comment. We have compared of our cgMLST scheme and MLST in lines 256 to 273. We performed a comparison analysis between cgMLST cluster analysis and SNP-based phylogeny (lines 274 to

284). Furthermore, we defined an outbreak threshold based on 7 allelic differences to applied our cgMLST scheme to hospital transmission investigations (lines 317 to 368). We performed an association analysis of CG and antibiotic resistance phenotypes (lines 402 to 421). We have added the association of resistance genes and virulence genes with wet-lab work in the Discussion section (lines 513 to 515 and lines 529 to 533).

There is a general lack of citations for some claims in the introduction, e.g. lines 61-65, 81-88

Response: Thank you for your advice. we have added the corresponding references in the revision (lines 82 to 90 and lines 105 to 109).

There should be more quantitative metrics of genome quality. What were the contamination and completeness %, number of contigs, and N50 values for each genome? Fragmented genomes will certainly affect the ability to annotate the genes appropriately, which would impact what is identified as core. Using genomes with N50s below, say, 25kb or over 300 contigs could have pretty significant impacts.

Response: Thank you for your advice. we have added information of the contamination and completeness %, number of contigs, and N50 values for each genome to Supplementary Table 1. The N50 values of all genomes are greater than 25kb, and the number of contigs of all strains are less than 300.

Note: During the analysis, we found that the isolate B73 had a contamination rate of 10.51%, which exceeded the contamination threshold 5%. Therefore, in our revised manuscript, the isolate B73 was not included in the validation step of the cgMLST scheme of *C. striatus*.

The identity and coverage thresholds for abricate are too low. Values in the range of 90-95% are more in line with current research when it comes to individual genes.

Response: Thank you for your advice. We have changed the identity and coverage

thresholds to 90% respectively and modified the corresponding Results section (lines 369 to 434).

Some of the claims regarding geographic barriers of transmission should be tempered and heavily caveated, e.g. lines 210-213. It is notable that there is no overlap between countries. However, the total number of isolates from outside of China is incredibly small and certainly does not represent the diversity that exists. If the authors want to try and better support this claim, they could try to look at the distribution of CGs within China to see if there is a lack of mixing of CGs in regions in the country.

Response: Thank you for your reminder. We have revised the claims regarding geographic barriers of transmission and explored the distribution of CGs within China. We have looked at the distribution of CGs within China to see the mixing of CGs in regions in the country (lines 241 to 255 and lines 463 to 475).

Minor comments:

Line 32: please change to "known diversity of this species"

Response: Thank you for your advice. We have rewritten this sentence as follows:

However, this method has not been reported in studies of *C. striatum*. In this work, we aim to propose a cgMLST scheme for *C. striatum*. All publicly available *C. striatum* genomes, 30 *C. striatum* strains isolated from the same hospital, and 1 epidemiologically unrelated outgroup *C. striatum* strain were used to establish a cgMLST scheme targeting 1,795 genes (hereinafter referred to as 1,795-cgMLST) (lines 51 to 55).

Lines 31, 38-39, 94, 347-349, etc: As mentioned previously, "outbreak" is not an appropriate word here until better defined.

Response: Thank you for your advice. We apologize for the improper understanding of the outbreak in the initially submitted manuscript. We have changed "outbreak" to

"hospital transmission" in the revised manuscript.

Line 42: Since there is no phenotypic or clinical outcome data, these CGs are "potentially hypervirulent and multidrug-resistant"

Response: Thank you for your advice. We have changed "potentially hypervirulent and multidrug-resistant" to "hypervirulent and multidrug-resistant " (line 64).

Lines 62 and 65: please provide examples of the "specific circumstances" and "range of diseases" with citations.

Response: Thank you for your advice. We rewrote the sentence as follows:

Corynebacterium striatum is part of microbiota of skin and nasal mucous membranes of humans and has been growingly reported as the etiologic agent of hospital and community acquired infections (1). Although the significance and prevalence of *C. striatum* as a causative pathogen are not yet fully understood, an increasing number of *C. striatum* isolates have been shown to be associated with a wide range of diseases, including septicæmias (2), pulmonary infection (3), meningitis (4), endocarditis (5), osteomyelitis (6), septic arthritis (7), skin wounds (8), and intrauterine infections (9) et al. Timely diagnosis and prompt treatment of the infection could lead to the favourable outcome of the patient (10) (lines 82 to 90).

Lines 87-88: please cite cgMLST here.

Response: Thank you for your advice. we have added the corresponding references in the revision (lines 111 to 112).

Line 91: do the authors mean "combining?"

Response: Thank you for your reminder. We rewrote the sentence as follows: The goal of this study was to develop a cgMLST scheme and delineate precisely, based on a cgMLST strategy, CGs corresponding to highly virulent and multidrug-resistant *C. striatum* isolates and investigate potential hospital transmission of *C. striatum* (lines 115 to 118).

Lines 228-233: There are many cases when it seems "isolate" should be singular, correct? It would help to better describe that CS refers to an isolate. Or better yet, the authors could remove CS altogether and refer to isolates with just a number. As an example: "With the exception of isolate 34 from patient 23..."

Response: Thank you for your advice. We have removed CS altogether and referred to isolates with just a number based on your example in the revised manuscript.

Lines 341-343: It is important to note that *C. striatum* isolates could also be re-introduced into these hospital units from environmental sources as well.

Response: Thank you for your advice. We have added the statements about that *C. striatum* isolates could also be re-introduced into these hospital units from environmental sources as well in lines 500 to 501.

Lines 354-359: These are bold claims that have not been supported by the evidence presented. These should either be removed or heavily caveated by the fact that the authors did not find anything to support this.

Response: Thank you for your advice. We have rewritten these claims and added related reference in lines 498 to 501.

November 17, 2022

Dr. Yutong Kang
Wenzhou Medical University
Wenzhou
China

Re: Spectrum01490-22R1 (Epidemiological investigation of hospital transmission of *Corynebacterium striatum* Infection by Core Genome Multilocus Sequence Typing approach)

Dear Dr. Yutong Kang:

Your manuscript has been accepted, and I am forwarding it to the ASM Journals Department for publication. You will be notified when your proofs are ready to be viewed.

Sincerely,

Daria Van Tyne
Editor, Microbiology Spectrum

Journals Department
Supplemental Material: Accept
Supplemental Material: Accept